# Unmasking coupling between channel gating and ion permeation in the muscle nicotinic receptor

John R Strikwerda[1], Steven M Sine[1,2,3]*

[1]Receptor Biology Laboratory, Department of Physiology and Biomedical Engineering, Rochester, United States; [2]Department of Molecular Pharmacology and Experimental Therapeutics, Rochester, United States; [3]Department of Neurology, Mayo Clinic College of Medicine, Rochester, United States

**Abstract** Whether ion channel gating is independent of ion permeation has been an enduring, unresolved question. Here, applying single channel recording to the archetypal muscle nicotinic receptor, we unmask coupling between channel gating and ion permeation by structural perturbation of a conserved intramembrane salt bridge. A charge-neutralizing mutation suppresses channel gating, reduces unitary current amplitude, and increases fluctuations of the open channel current. Power spectra of the current fluctuations exhibit low- and high-frequency Lorentzian components, which increase in charge-neutralized mutant receptors. After aligning channel openings and closings at the time of transition, the average unitary current exhibits asymmetric relaxations just after channel opening and before channel closing. A theory in which structural motions contribute jointly to channel gating and ion conduction describes both the power spectrum and the current relaxations. Coupling manifests as a transient increase in the open channel current upon channel opening and a decrease upon channel closing.

*For correspondence: sine@mayo.edu

Competing interests: The authors declare that no competing interests exist.

## Introduction

The superfamily of pentameric ligand-gated ion channels (pLGICs) mediates rapid excitatory and inhibitory signaling throughout the central and peripheral nervous systems. The conserved five sub-unit architecture enables transduction of the free energy of neurotransmitter binding into gating of an intrinsic ion channel and rapid flow of ions down their electrochemical gradients. The body of work over several decades has revealed a modular design imbedded within the pentameric architecture, including specialized structures that form the neurotransmitter binding sites, the ion channel formed by hydrophobic α-helices and the ion selectivity filter (*Nemecz et al., 2016*; *Sine and Engel, 2006*). In addition, structural bases for coupling among these modules have started to emerge. For instance, a series of intermolecular interactions among conserved residues at strategic locations within the tertiary structure are essential in coupling neurotransmitter binding to channel gating, a signature function of pLGICs (*Mukhtasimova et al., 2005*; *Mukhtasimova and Sine, 2007*; *Mukhtasimova and Sine, 2013*; *Lee and Sine, 2005*). However, whether gating of the channel and ion flow through the channel are independent processes has remained elusive.

To investigate whether channel gating and ion conduction are coupled, we took advantage of the increasing library of high-resolution structures of members of the pLGIC superfamily. Within all eukaryotic pLGIC structures solved to date, each subunit contains a salt bridge formed by a pair of conserved basic and acidic residues (*Figure 1*). The basic residue is located at the 0' position of the M2 α-helix that lines the pore and is adjacent to the residue that forms the ion selectivity filter, a glutamate in cation-selective channels and a nonpolar residue in anion-selective channels. The acidic residue stems from the outermost M4 α-helix and contacts the basic residue within the four-helix

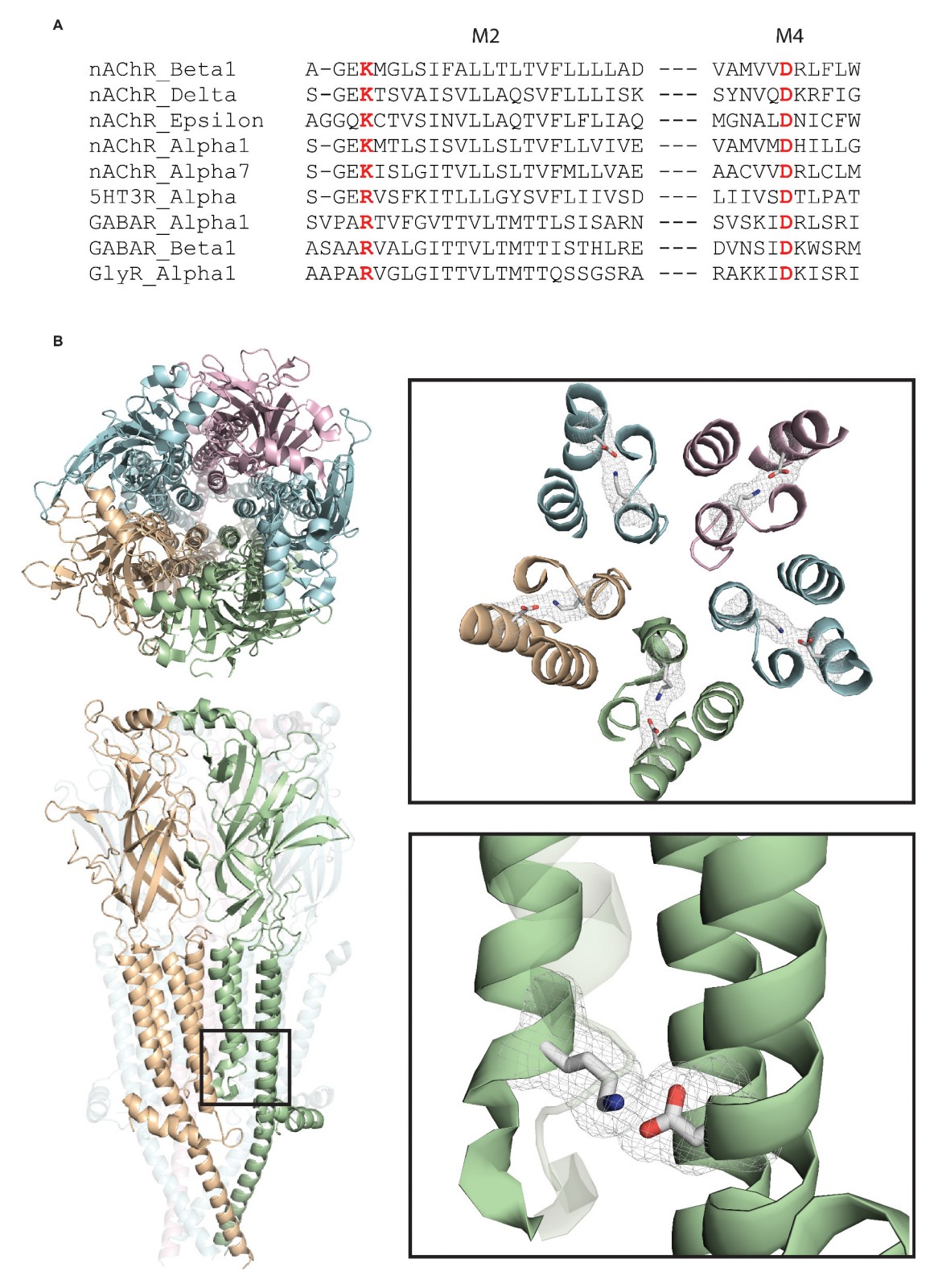

**Figure 1.** A conserved intra-subunit salt bridge links pore-lining and peripheral α-helices. (**A**) Sequence alignment human of pLGIC subunit M2 and M4 domains. (**B**) Structure of the *Torpedo* receptor (PDB: 6UWZ) viewed perpendicular (upper) and parallel (lower) to the cell membrane. Close up views show the salt bridge between the M2 and M4 α-helices rendered as sticks with surface mesh overlaid.

transmembrane bundle. The presence of this interaction in all eukaryotic pLGICs suggests it is important for receptor function. Indeed, recent studies showed that altering this interaction reduced or eliminated agonist-elicited macroscopic currents in homomeric 5-HT$_3$ and α7 receptors (*Mesoy et al., 2019*; *da Costa Couto et al., 2020*), suggesting it contributes to subunit folding or assembly. On the other hand, the proximity of the salt bridge to the region of the pore that forms the ion selectivity filter suggests contributions to channel gating, ion permeation, or both.

To investigate the functional contribution of the conserved salt bridge, we studied the archetypal muscle nicotinic acetylcholine receptor (AChR) that mediates neuromuscular synaptic transmission. The muscle AChR offers the advantage that functional measurements at the level of single receptor channels have been well established, including the speed and efficiency of channel gating and conduction of ions through the channel (*Mukhtasimova et al., 2016*; *Gardner et al., 1984*). In addition, a high-resolution structure of the physiologically analogous *Torpedo* AChR was determined recently (*Rahman et al., 2020*). Herein, we combine structural perturbations of the conserved salt bridge with single channel measurements of channel gating, fluctuations in the current through the open channel, and relaxations of the unitary current upon channel opening and closing. The results reveal that the conserved salt bridge contributes to channel gating, ion conduction, and coupling between the two processes.

## Results

### Receptor activation at the single channel level

To test the contribution of the pore-peripheral salt bridge toward activation of the muscle receptor, we disrupted the interaction via mutagenesis, expressed cDNAs encoding mutant and complementary wild-type receptor subunits in clonal mammalian fibroblasts, and studied the resultant receptors using single channel patch clamp electrophysiology. Alterations to the 0′ lysine residue alone were not well tolerated and therefore prohibited study, consistent with previous studies (*Cymes and Grosman, 2011*). However, mutating the aspartate residue on the peripheral M4 α-helix to an asparagine was tolerated. This aspartate to asparagine mutation (DN) represents the most conservative change possible as it retains the chain length of the native aspartate residue while neutralizing the aspartate's acidic group. Thus, the DN mutation was used throughout this work to disrupt the salt bridge.

*Figure 2* summarizes the impact of disrupting the salt bridge on ACh-elicited single channel currents. In panel A, exemplar single channel currents and dwell time histograms are shown for the wild-type muscle receptor and for receptors harboring the DN mutation in either the δ- or β-subunit, that is βD445N and δD449N. Receptors with the DN mutation exhibit briefer open channel dwell times and longer closed channel dwell times compared to their wild-type counterpart. However, the magnitudes of these effects are not equivalent between the δ- and β-DN mutants. Namely, channel dwell times are most impacted in the β-DN compared to the δ-DN mutant. Further, minimal changes in open and closed dwell times were observed when the DN mutation was placed in either the α- or ε-subunit (*Figure 2—figure supplement 1*). The present work investigates the DN mutation in the β- and δ-subunits, while a subsequent paper will investigate the subunit dependence of the salt bridge interaction. Notably, in addition to reducing the overall length of open dwell times, the DN mutation gives rise to a second exponential component of openings that persists even in the presence of saturating concentrations of ACh. This second component corresponds to an additional open state that is presumably doubly bound to agonist, as it is observed at high ACh concentrations. Relative areas and mean dwell times for currents elicited by each ACh concentration are given in *Supplementary file 1 c-e*.

The changes in open and closed dwell times observed in the DN mutant receptors indicate a change in the kinetics of receptor activation. However, from these dwell time data alone, we cannot identify the elementary kinetic step altered by the mutation. Namely, the observed changes in single channel dwell times could result from impaired ACh binding, impaired channel gating, or a combination of the two. In addition, receptors harboring the DN mutation exhibit a second open state that persists at high agonist concentrations, thus a single mechanism cannot describe the dwell time distributions for both the mutant and wild-type receptors. However, general insight into the altered kinetic step, agonist binding or channel gating, can be obtained from changes in the channel open

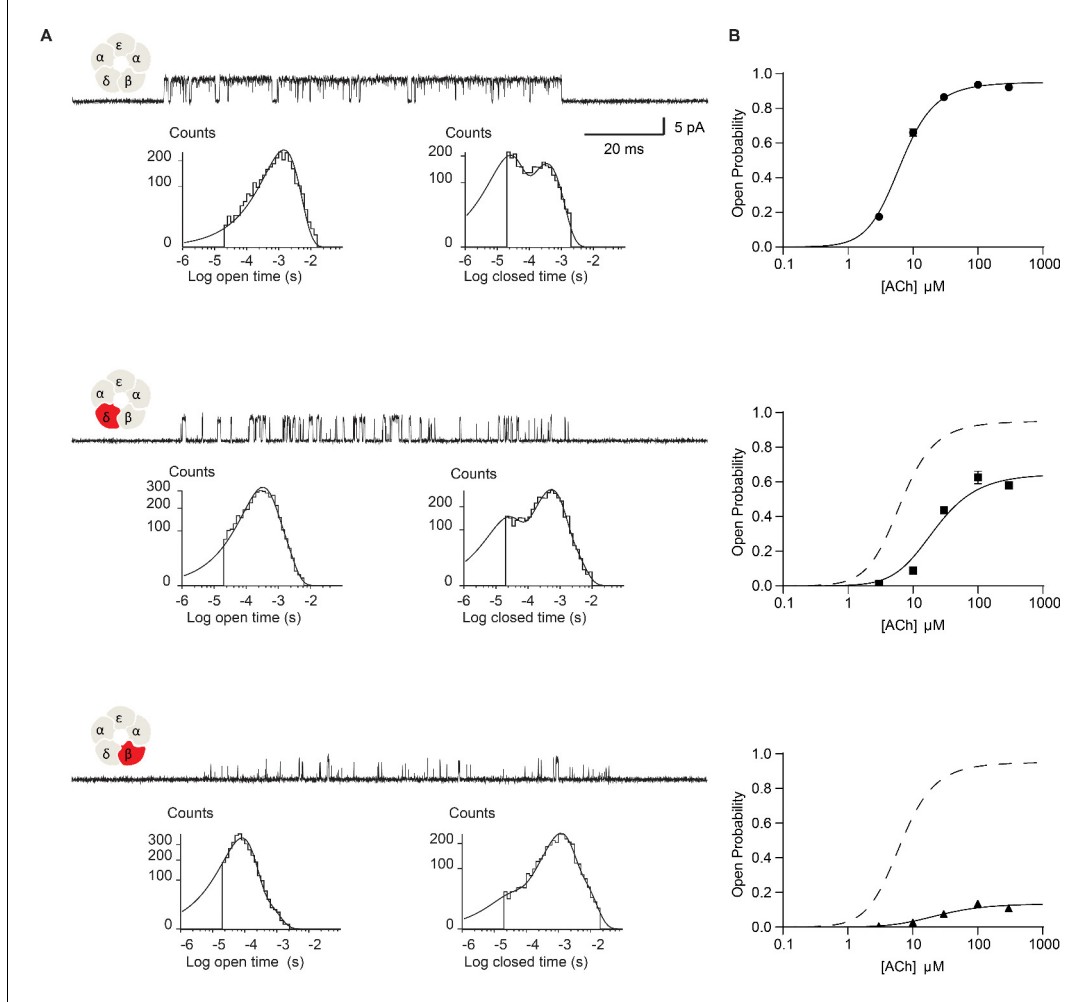

**Figure 2.** Disrupting the salt bridge impairs channel gating efficiency. (**A**) Single channel currents and dwell time histograms from wild-type, β-DN, and δ-DN receptors recorded in the presence of 30 μM ACh and with an applied membrane potential of −70 mV. Channel openings are upward deflections from baseline. For each type of receptor, a cluster of openings from a single receptor channel is shown at a bandwidth of 10 kHz, along with dwell time histograms fitted by the sum of exponentials. Histograms are from a representative patch. Mean fits are given in *Supplementary file 1c-e*. (**B**) Corresponding plots of channel probability versus ACh concentration fitted by a simple sequential bind-bind-gate model with response function $\frac{\theta}{1+\theta+\frac{2K}{[ACh]}+\frac{K^2}{[ACh]^2}}$. Points represent the average open probability within clusters from three independent patches displayed with 95% confidence intervals.

The online version of this article includes the following source data and figure supplement(s) for figure 2:

**Source data 1.** Open probability data for plots in *Figure 2B*.

**Figure supplement 1.** Single channel dwell times and open probabilities of the α-DN and ε-DN receptor remain similar to wild type.

probability as a function of ACh concentration. Changes in agonist binding will manifest as a lateral shift of the open probability versus ACh concentration with no change in the maximum open probability. Changes in channel gating will manifest as a change in the maximum open probability with a concomitant lateral shift of the open probability versus ACh concentration (*Colquhoun, 1998*).

*Figure 2B* shows the dependence of the channel open probability on ACh concentration. Profiles for the δ- and β-DN mutants exhibit reductions in the maximum open probability and are right-shifted, consistent with a reduction in the gating efficiency of the receptor. To gain further insight into mechanism, we fit a simple sequential bind-bind-gate mechanism, $R \overset{K}{\leftrightarrow} RA \overset{K}{\leftrightarrow} RAA \overset{\theta}{\leftrightarrow} RAA^*$, to the open probability data. The analysis reveals a large effect of the salt bridge disrupting mutation on the gating isomerization step. Specifically, $\theta$ is reduced from 19.2 in the wild-type receptor (95% CI = 17.5 to 21.2) to 1.86 in the δ-mutant (95% CI 1.70 to 1.95) and 0.154 in the β-mutant (95% CI 0.147 to 0.162), representing over 10- and 100-fold reductions, respectively. In contrast, the

dissociation constants for the mutants remain close to the WT value, but show a slight trend toward higher affinity, taking values of 23.14 µM in the wild-type receptor (95% CI = 21.6 µM to 24.86 µM), 20.28 µM in the δ-mutant (95% CI = 18.25 µM to 22.52 µM), and 11.15 µM in the β-mutant (95% CI = 9.76 µM to 12.73 µM). These results localize the primary effect of the salt bridge disrupting mutation to the terminal step in receptor activation: channel gating. In further support of an effect on channel gating, the mutations destabilize the open state, as shown by decreases in the average open channel dwell time. Hence, we conclude that in the wild-type receptor the salt bridge facilitates the terminal step of receptor activation in which the agonist-bound closed receptor transitions between fully occupied closed and open states.

To investigate whether the effect on channel gating depends upon the number of mutant subunits, we co-transfected cells with both the β-DN and δ-DN mutant subunits and studied the resultant receptors. Single channel currents were again recorded over a range of ACh concentrations and the channel open probability determined. Relative to receptors harboring the β-DN mutant subunit alone, receptors with the β + δ DN mutant show a diminished maximum open probability, a reduction in the mean open dwell time, and prolongation of the closed dwell times (*Figure 3*). Thus, when β- and δ-DN mutant subunits are combined, channel gating efficiency decreases beyond that observed with either single mutant, demonstrating a mutant dose dependence impacting receptor channel gating.

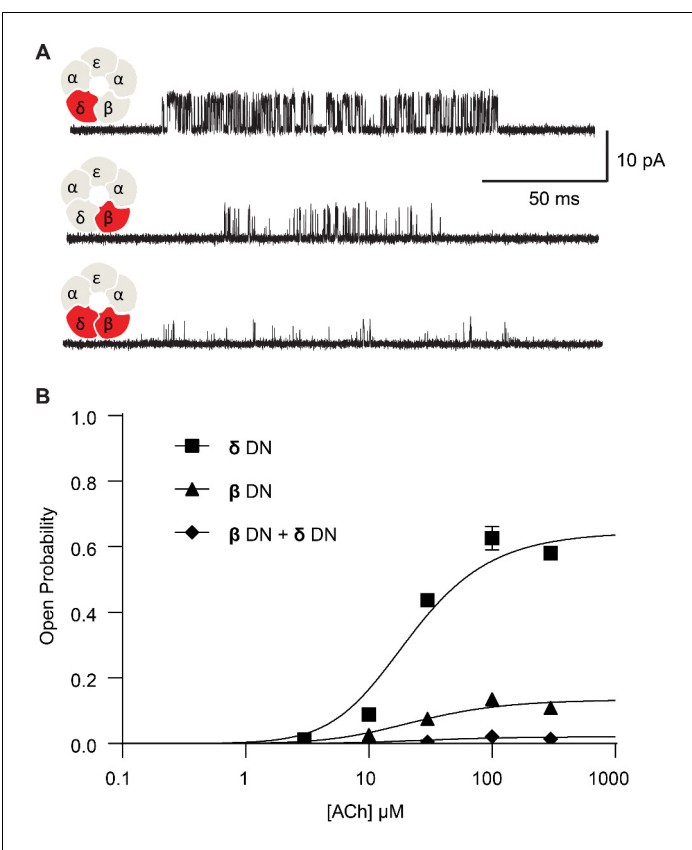

**Figure 3.** Channel gating efficiency depends on the number of mutant subunits. (**A**) Single channel currents recorded from single and double DN mutant receptors in the presence of 300 µM ACh with applied membrane potential of −70 mV and bandwidth of 10 kHz. (**B**) Corresponding plots of channel open probability versus ACh concentration fitted by a simple sequential bind-bind-gate model with response function $\frac{\theta}{1+\theta+\frac{2K}{[ACh]}+\frac{K^2}{[ACh]^2}}$. Points represent the average open probability within clusters from three independent patches displayed with 95% confidence intervals.

The online version of this article includes the following source data for figure 3:

**Source data 1.** Open probability data for plots in *Figure 3B*.

Next, we hypothesized that if breaking the salt bridge impairs channel gating, reversing the charges on the pair of interacting residues, thereby restoring the electrostatic interaction, would restore efficient channel gating. Therefore, we engineered β- and δ- subunits with a positively charged lysine residue on the peripheral-most M4 α-helix and a negatively charged aspartate residue on the pore-lining M2 α-helix. *Figure 4* summarizes the results from recordings of ACh-elicited single channel currents for these constructs. Exemplar single channel currents and channel open probability profiles for receptors harboring the salt bridge disrupting mutation are compared to those in which the positions of the charged residues have been exchanged. Single channel open times increase for the charge-exchanged compared to the charge-neutralized receptors, and similarly, the closed times decrease. Furthermore, the maximum open probability for the charge-exchanged receptors markedly increases, and the gating equilibrium constants rise from 1.86 to 9.59 (95% CI 8.254 to 11.34) and 0.154 to 3.153 (95% CI 2.93 to 3.40) for charge-neutralized (DN) to charge-exchanged (DKKD) δ- and β-subunits, respectively. Thus, gating in the charge exchanged receptors is significantly enhanced, although it does not return to that of the wild-type receptor. The fact that gating does not return entirely to that of wild type is perhaps expected given that the local environment around charge exchanged residues differs from that of the native salt bridge interaction. Nonetheless, the enhanced channel gating in the charge-exchanged receptors suggests that it is not simply the positions of the residues that form the salt bridge but rather the electrostatic interaction between the residues that is critical for efficient channel gating.

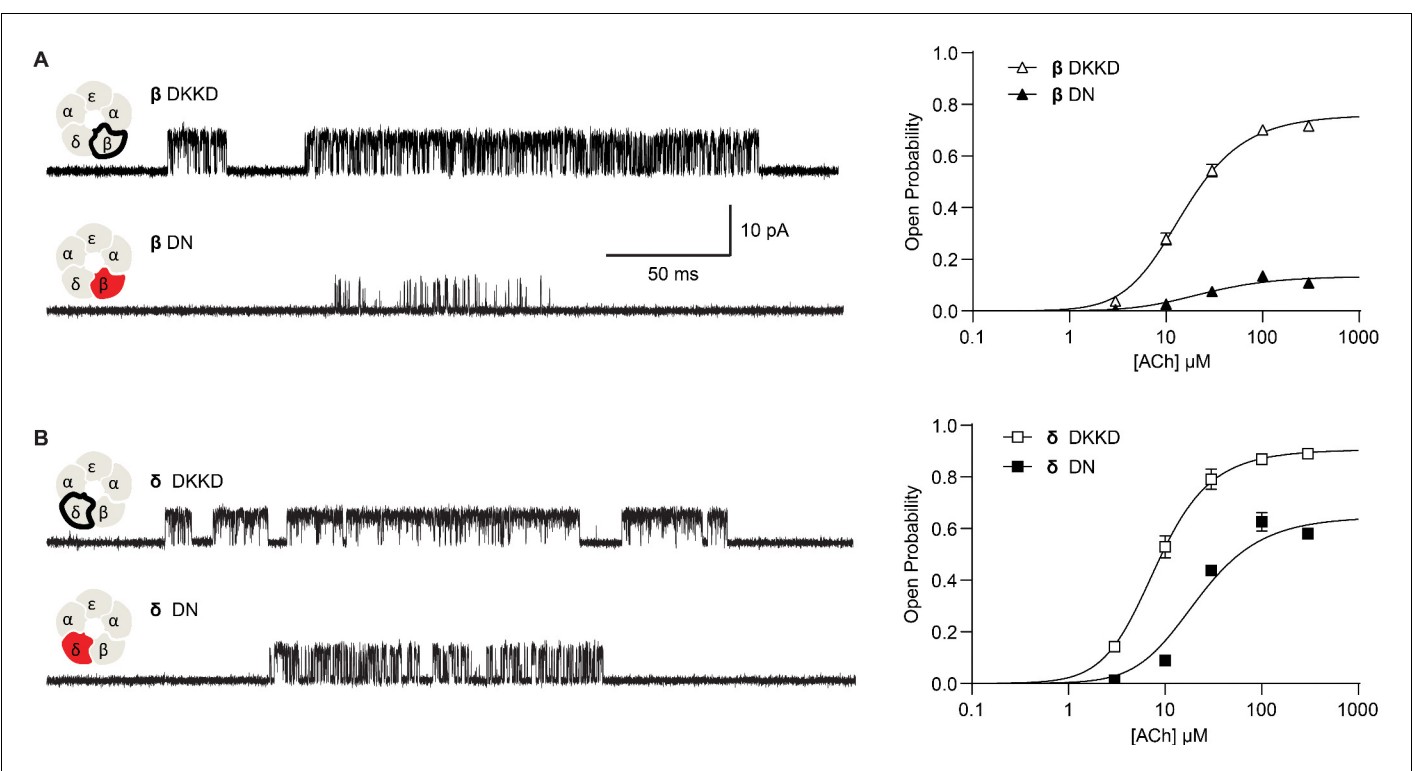

**Figure 4.** Reversing the charged residues of the salt bridge recovers wild-type-like channel gating. (**A**) Single channel currents recorded from charge neutralized (DN) and charge reversed (DKKD) receptors in the presence of 300 μM ACh with an applied membrane potential of −70 mV and bandwidth of 10 kHz. (**B**) Corresponding plots of channel open probability versus ACh concentration fitted by a simple sequential bind-bind-gate model with response function $\frac{\theta}{1+\theta+\frac{2K}{[ACh]}+\frac{K^2}{[ACh]^2}}$. Points represent the average open probability within clusters from three independent patches displayed with 95% confidence intervals.

The online version of this article includes the following source data for figure 4:

**Source data 1.** Open probability data for plots in *Figure 3A*.
**Source data 2.** Open probability data for plots in *Figure 4B*.

## Ion permeation

Owing to the proximity of the salt bridge to the region of the pore that forms the ion selectivity filter, we hypothesized that the DN mutation may also alter the flow of ions through the channel. *Figure 5* shows close up views of single channel currents for wild-type receptors and receptors harboring the DN mutation in both the β- and δ-subunits. All points histograms generated from the digitized current traces are displayed alongside. The histograms show a reduced unitary current amplitude as well as a broadening of the open channel current envelope in mutant compared to wild-type receptors. While these data demonstrate that the DN mutation impacts ion conduction, including an increase in fluctuations of the open channel current, the brief open times of the mutant receptors prohibited detailed study. Therefore, to better quantify changes in open channel current fluctuations, we installed a previously characterized open time prolonging mutation in the ε subunit (*Ohno et al., 1995*). Originally discovered as the cause of a slow channel congenital myasthenic syndrome, the ε T264P mutation is located at the 12' position of M2, and prolongs single channel open and burst durations via a decrease in the channel closing rate. The mutation does not affect the unitary current amplitude and therefore has been applied as a tool for studying single channel conducting properties of the transient open state of the muscle receptor (*Cymes and Grosman, 2011*; *Cymes and Grosman, 2012*). For clarity, in the following sections we refer to receptors that harbor the ε T264P mutation as either salt bridge intact or DN receptors.

Representative traces for receptors with and without the DN mutation in the β- and δ-subunits are shown in *Figure 6A*. For these studies we increased the membrane potential from −70 mV to −120 mV to better resolve the open channel current. *Figure 6A* shows a subtle but appreciable broadening of the open channel noise envelope for the β-DN mutant relative to the salt bridge intact receptor that is further increased when combined with the δ-DN mutant. This mutant dose-dependent increase in open channel current fluctuations mirrors the dose-dependent suppression of channel gating efficiency shown in *Figure 3*. All points histograms of the open channel current for the salt bridge intact receptor are compared to those for the β-DN and the β + δ DN mutants in *Figure 6B*. Here, we see clearly a progressive broadening of the open channel current envelope that is quantified in *Figure 6C*.

To probe the frequency content of the open channel current fluctuations, we constructed power spectra of the open channel and baseline currents and display the difference spectra in *Figure 6D*. In each case, the spectra can be fit by the sum of either one or two Lorentzian components plus a flat frequency-independent component.

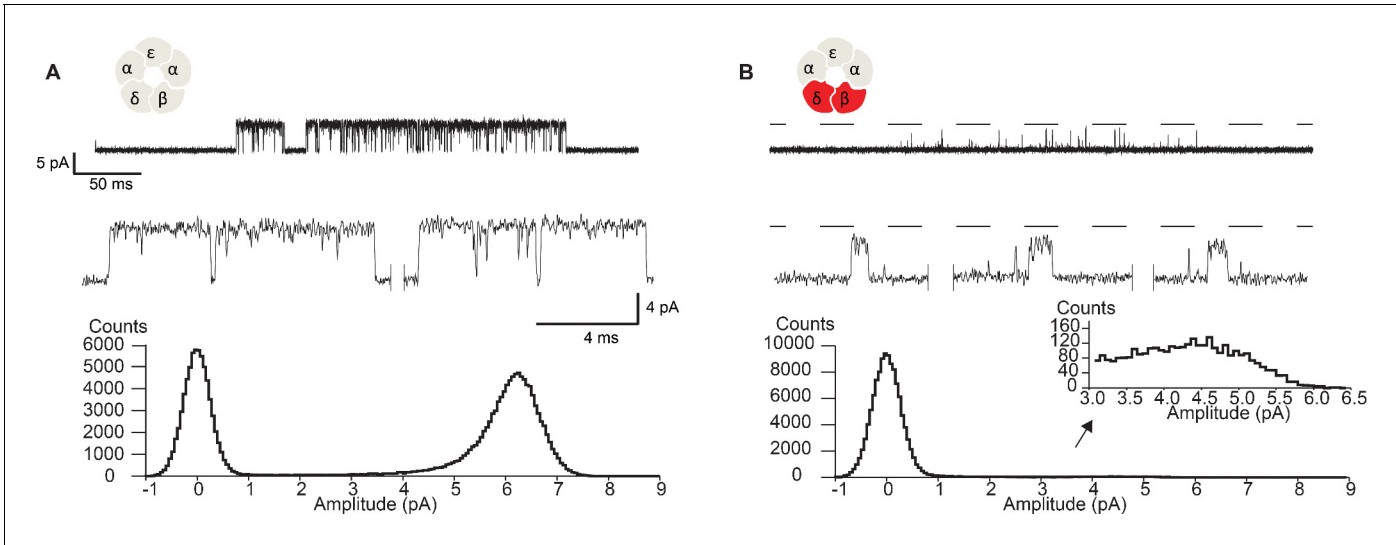

**Figure 5.** Disrupting the salt bridge diminishes the unitary current amplitude and increases open channel fluctuations. Single channel currents and histograms of the digitized points from wild-type (**A**) and β-DN + δ-DN (**B**) receptors recorded in the presence of 30 µM ACh with an applied membrane potential of −70 mV and bandwidth of 10 kHz. Dashed line corresponds to the wild-type unitary current amplitude.

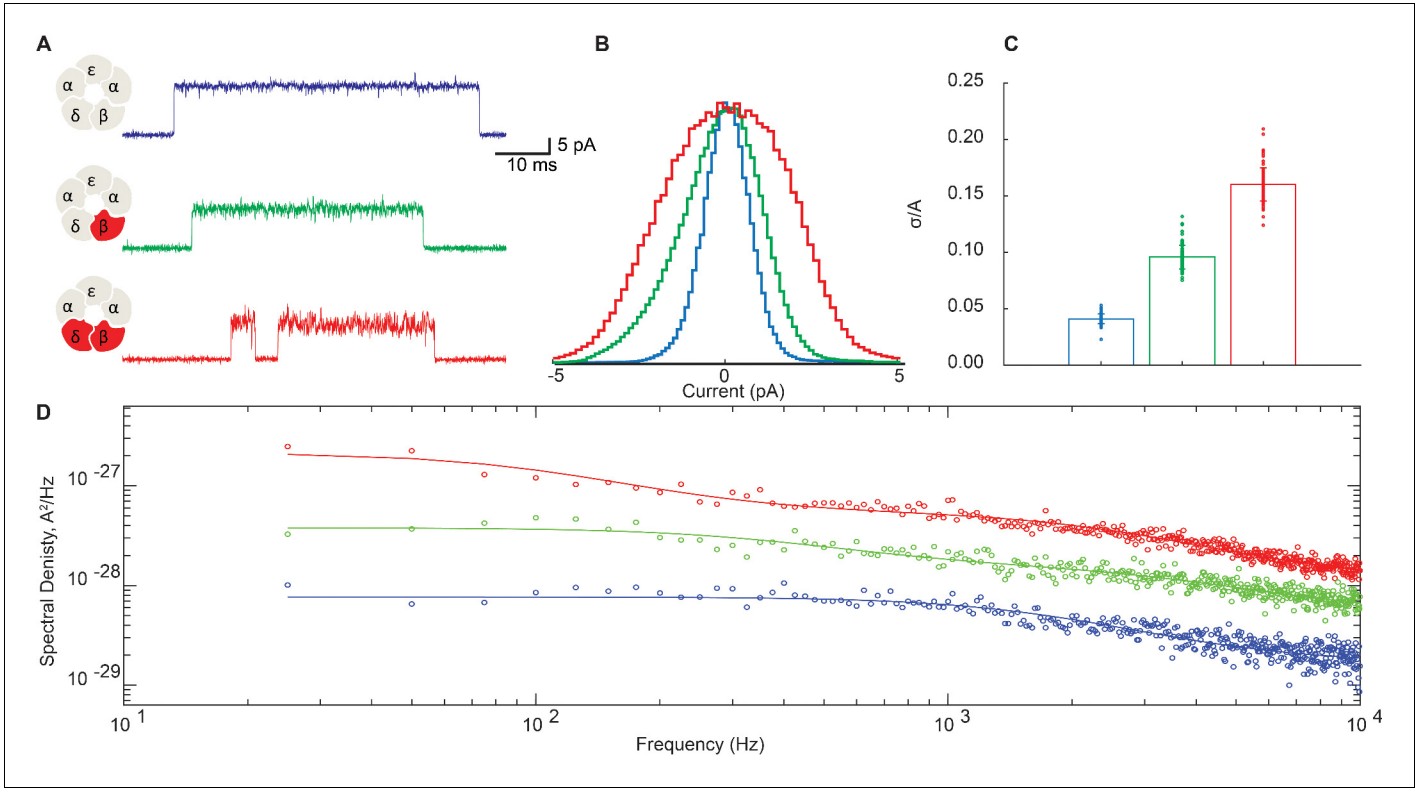

**Figure 6.** Disrupting the salt bridge increases both low- and high-frequency open channel current fluctuations. (**A**) Representative channel openings from salt bridge intact and salt bridge disrupted DN receptors harboring the open time prolonging ε-T264P mutation in the presence of 100 nM ACh with a membrane potential of −120 mV and bandwidth of 10 kHz. (**B**) Corresponding histograms of open channel current constructed from digitized points from at least 50 open channel current stretches, displayed centered on the mode value. (**C**) Standard deviation of the open channel current relative to the open channel current amplitude. (**D**) Open channel power spectra computed as the average difference between open channel spectra and flanking baseline spectra calibrated according to the frequency response of the recording system (*Figure 6—figure supplement 1*). Each spectrum is fit by either a one or two component Lorentzian function plus a frequency-independent component.

The online version of this article includes the following source data and figure supplement(s) for figure 6:

**Source data 1.** Open channel current values for histograms plotted in *Figure 6B*.
**Source data 2.** Open channel standard deviation relative to channel amplitude plotted in *Figure 6C*.
**Figure supplement 1.** Frequency response of the recording system.

$$S(f) = \frac{S_S}{1 + \left(\frac{f}{fc_s}\right)^2} + \frac{S_f}{1 + \left(\frac{f}{fc_f}\right)^2} + S_c. \tag{1}$$

Magnitudes (S) and cut off frequencies ($f_c$) for each Lorentzian component and the magnitude of the frequency-independent component ($S_c$) are tabulated in *Supplementary file 2a*. For the salt bridge intact receptor, the power spectrum is best described by a single Lorentzian component, with a relatively fast cut off frequency of around 2kHz, plus a frequency independent component. Relative to the salt bridge intact receptor, the DN mutant receptors show a marked increase in spectral density across all frequencies and are best fit with two Lorentzian components plus a frequency independent component. One Lorentzian component of the DN receptors shares a similar cut off frequency to that of the salt bridge intact receptor, whereas the other Lorentzian component is appreciably slower. Magnitudes for both the fast Lorentzian component and the frequency-independent component are markedly increased in the DN receptor. Thus, the DN mutation amplifies current fluctuations of a similar time course observed in the salt bridge intact receptor and also introduces a slower fluctuation process that is not detected in the salt bridge intact receptor.

## Coupling between open channel current fluctuations and channel gating-

The DN mutation represents the first mutation to our knowledge that affects both channel gating and fluctuations in the open channel current. In pioneering studies of open channel current fluctuations, Sigworth hypothesized that molecular motions that underlie fluctuations in the open channel current might be coupled to channel gating *Sigworth, 1986*. However, a test of this hypothesis applied to wild-type receptors in rat myocytes determined that coupling, if present, was too small to be detected. Because the DN mutation markedly increases fluctuations in the open channel current, and simultaneously affects channel gating, we sought to test directly for coupling between open channel current fluctuations and channel gating. Implementing methods analogous to those developed by Sigworth, we segmented many opening and closing transitions, aligned them at the time of the transition, and summed them to produce an average open channel current profile. If fluctuations in the open channel current are independent of the gating transition, the fluctuations will be randomly distributed at the time of the gating transition, and the average open channel current profile should appear as random fluctuations about a horizontal line. However, if fluctuations in the open

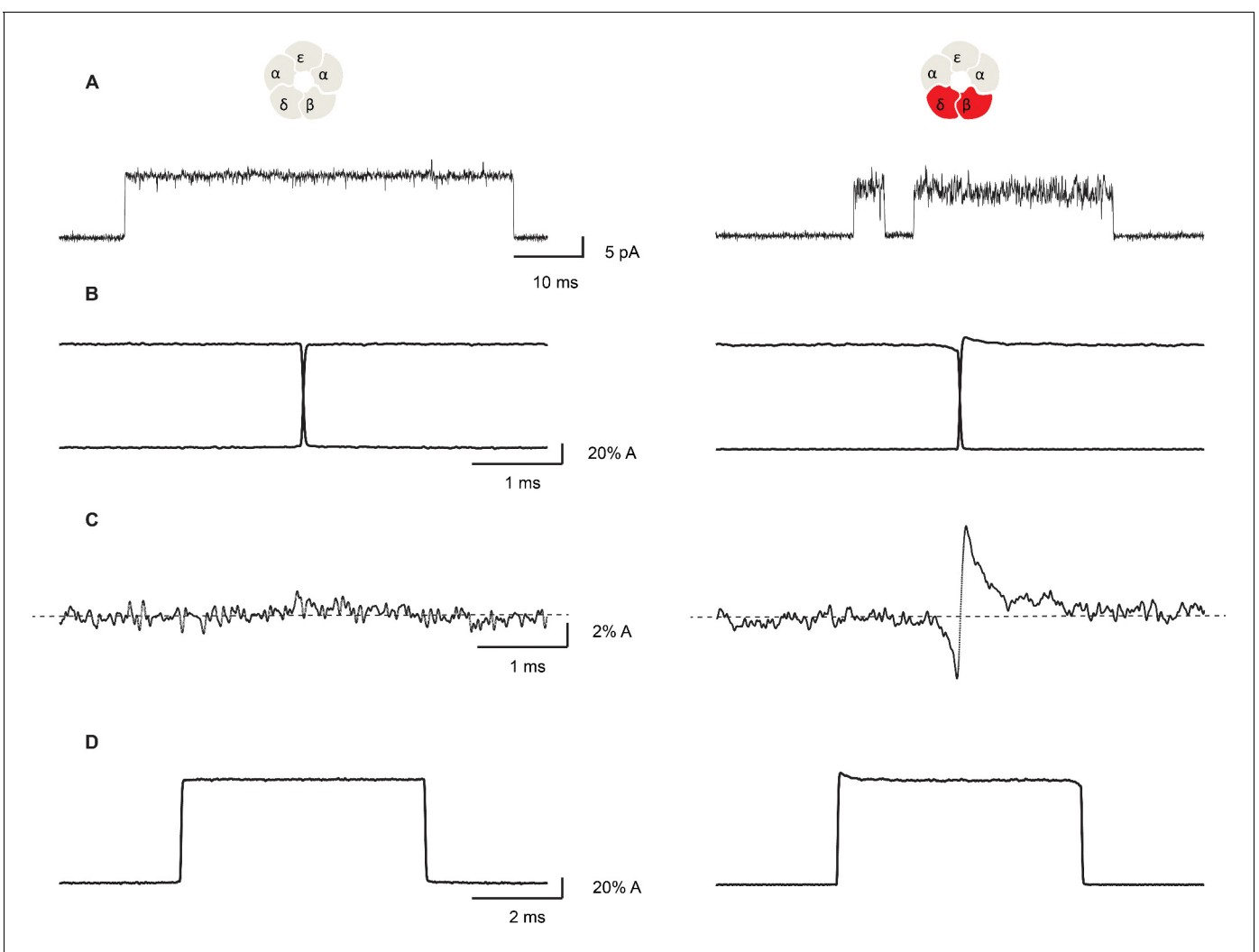

**Figure 7.** Open channel current fluctuations are coupled to channel gating in a salt bridge disrupted mutant receptor. Representative single channel openings (A), average channel opening and closing transitions aligned at their midpoints (B), the sum of the averaged openings and closings (C), and the average unitary current pulse (D). For the salt bridge intact receptor, 695 opening and 722 closing transitions were analyzed. For the DN mutant, 1000 opening and 1004 closing transitions were analyzed. Recordings were obtained in the presence of 100 nM ACh with a membrane potential of −120 mV and a bandwidth of 10 kHz.

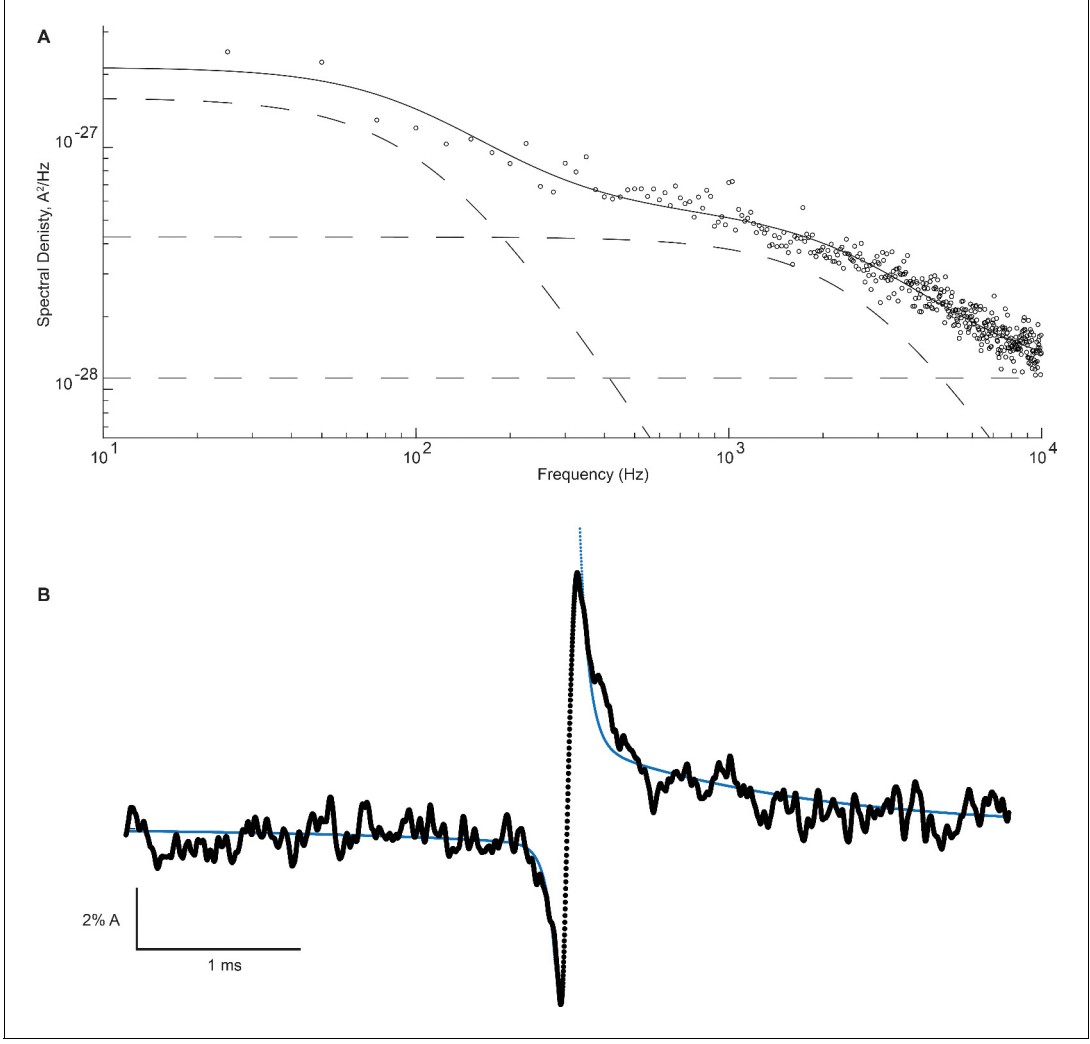

**Figure 8.** High- and low-frequency open channel current fluctuations are coupled to channel gating. (**A**) Open channel power spectrum computed for the β-DN + δ-DN receptor. Individual components of the spectrum are shown as dashed lines. (**B**) Corresponding sum of averaged opening and closing transitions, as in *Figure 7c*. Smooth curves overlaid on the power spectra and the on and off gating relaxations are results from a simultaneous fit of *Equations 1, A7, A8* to the respective data.

The online version of this article includes the following figure supplement(s) for figure 8:

**Figure supplement 1.** Correlation of slow and fast Lorentzian fluctuations required to fit experimental average transition profiles.

**Figure supplement 2.** Coupling of ion permeation and gating replicated in three independent recordings.

**Figure supplement 3.** Hypothetical on and off gating relaxations for the salt bridge intact receptor.

channel current are correlated with the gating transition, the average open channel current profile may show transient increases or decreases near the time of the transition.

*Figure 7* compares representative channel openings, averaged openings and closings aligned at the time of the gating transition, and their sum for the salt bridge intact and the β + δ DN mutant receptors. For the salt bridge intact receptor, the sum of the averaged openings and closings shows small fluctuations of the current about a horizontal line, with no discernable change at the time of the gating transition. In contrast, the β + δ DN mutant receptor shows transient relaxations of the current just after the channel opens and just before it closes. Specifically, we see that on average, the channel closes from a lower conducting state and opens to a higher conducting state, indicating that fluctuations that increase conductance are correlated with channel opening and fluctuations that decrease conductance are correlated with channel closing.

If the fluctuations in open channel current are coupled to channel gating, a further expectation is that the relaxations in the average opening and closing transitions will have time constants corresponding to the frequency of the correlated fluctuations. For example, if the low-frequency Lorentzian component of the open channel current fluctuations is correlated with the gating transition, slow exponential relaxations should be present within the averaged open channel current profile. Similarly, if the high-frequency Lorentzian component is correlated with the gating transitions, fast exponential relaxations should be present. And if both components are correlated with the gating transitions, biexponential relaxations should be present. Further, the amplitudes of the relaxations indicate the efficiency of coupling between the fluctuations in open channel current and the channel opening and closing transitions.

*Figure 8* shows the on and the off relaxations determined for the β + δ DN mutant receptor and the corresponding power spectrum of the open channel current fluctuations. Superimposed on each relaxation is a two-component exponential function with time constants determined by simultaneously fitting the power spectrum of the open channel current and the on and off relaxations; the time constant for exponential decay, $\tau$, for the relaxation is related to the cutoff frequency, $f_c$, in the power spectrum by $\tau = 2\pi f_c$. A one-component exponential function, using either the high or low-frequency Lorentzian cutoff frequency, produced a poor fit to either the on or the off relaxation (*Figure 8—figure supplement 1*). However, a two-component fit comprising both the low- and high-frequency fluctuations in the open channel current provide a good fit to the on and off relaxations, thus demonstrating that the two processes are functionally coupled. This close correspondence between the power spectrum of the open channel current and the on and off relaxations was replicated in recordings from three independent patches (*Figure 8—figure supplement 2*). Fitted parameters for the on and off relaxations and the power spectrum are tabulated in *Supplementary file 2b*. The analysis further reveals that the high-frequency fluctuations show relatively stronger coupling than the low-frequency fluctuations, and that the opening transition displays greater coupling compared to the closing transition. Thus, there is asymmetry in the coupling between the open channel current fluctuations and the channel gating transitions.

## Discussion

We demonstrate that a conserved structural motif, a pore-peripheral salt bridge spanning inner and outer transmembrane α-helices in the muscle AChR, contributes to both the open channel current and the transition rates governing channel opening and closing. For when the salt bridge is disrupted, we observe changes in the amplitude and variance of the single channel current accompanied by changes in single channel open and closed dwell times. Thus, ion flow through the channel and gating of the channel are coupled to a common structure. However,

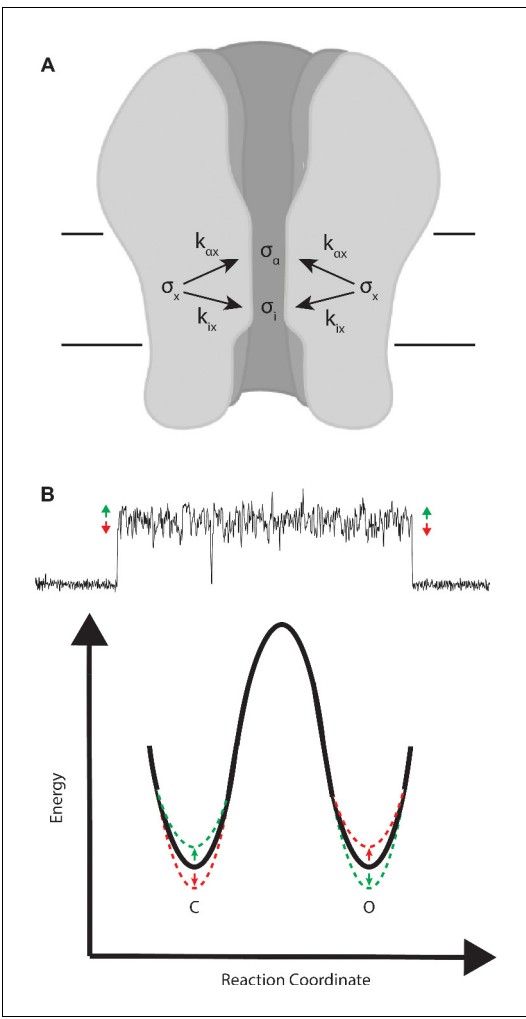

**Figure 9.** Mechanistic interpretation of coupling between ion flow and channel gating. (**A**) Schematic showing the impact of motion **x** on ion flow and channel gating. (**B**) Motions correlated with increased ion flow increase the rate of channel opening and slow the rate of channel closing (green). Motions correlated with reduced ion flow, slow the rate of channel opening, and increase the rate of channel closing (red).

despite our evidence for structural coupling, these observations alone do not distinguish whether the two processes are functionally coupled. Instead functional coupling is demonstrated by comparing power spectra of the fluctuations in open channel current with relaxations in the time course of the average unitary current. The results reveal that the fast and slow fluctuations of the open channel current give rise to transient relaxations in the time course of the average unitary current.

To account for functional coupling between ion flow and channel gating, as originally postulated by *Sigworth, 1986*, we envision an intramolecular motion *x* that jointly affects the open channel current *i* and the channel gating rate constant, say $\alpha$ for the channel closing transition (*Figure 9*). Motion *x* is a stochastic process, with standard deviation $\sigma_x$, which is coupled to both the open channel current and the transition rate constant, which are also stochastic processes with standard deviations $\sigma_i$ and $\sigma_\alpha$, respectively (Appendix, Part 1). Coupling between motion *x* and either *i* or $\alpha$ is described by the coupling coefficients, $k_{ix}$ and $k_{\alpha x}$, which relate fluctuations in *i* and $\alpha$ to fluctuations in motion *x* according to

$$\sigma_i = k_{ix}\sigma_x \tag{2}$$

$$\sigma_\alpha = k_{\alpha x}\sigma_x \tag{3}$$

Thus, motion *x* effects both the ionic current and channel gating as determined by the magnitudes of the coupling coefficients.

Our results demonstrate that fluctuations in the open channel current quantitatively describe the on and off relaxations in the average unitary current. Strikingly, these relaxations are clearly present in the DN mutant but not in the salt bridge intact receptor, raising the question of what differs mechanistically between the two types of receptors. To elicit a change in the amplitude of the on and off relaxations, the mutation could alter one of the three elementary parameters defined above: $\sigma_x$, $k_{ix}$, or $k_{\alpha x}$. From these three possible scenarios, we can exclude the third one because a change in $k_{\alpha x}$ alone cannot explain the increase in open channel current fluctuations observed experimentally.

To gain insight into which of the remaining parameters, $\sigma_x$ or $k_{ix}$, could account for the change in relaxation amplitude and open channel current fluctuations, we calculate hypothetical relaxation amplitudes for the salt bridge intact receptor under the assumption that the mutation effects either $\sigma_x$ or $k_{ix}$ (*Figure 8—figure supplement 3*). For the case in which the mutation elicits a change in $\sigma_x$, the expression relating the salt bridge intact relaxation amplitude to the DN mutant relaxation amplitude is given by (Appendix, part 2):

$$A_{wt} = \left(\frac{\sigma_{iwt}}{\sigma_{iDN}}\right)^2 A_{DN}. \tag{4}$$

Alternatively, for the case in which the mutation changes $k_{ix}$, the expression becomes

$$A_{wt} = \left(\frac{\sigma_{iwt}}{\sigma_{iDN}}\right) A_{DN}. \tag{5}$$

Amplitudes calculated according to either *Equations 4 and 5* are within the experimental noise of the average open current of the salt bridge intact receptor (*Figure 8—figure supplement 3*). Therefore, our data do not allow us to distinguish between these scenarios. However, because the mutation affects both the ionic current and channel gating, we favor the scenario in which the mutation increases $\sigma_x$. This scenario is consistent with the dual effect of the mutation on fluctuations in the open channel current and rate constants for channel gating. In either case our data suggest that in the salt bridge intact receptor, fluctuations in the open channel current may be coupled to channel gating, but the fluctuations are not large enough to give rise to detectable on and off relaxations.

For the β + δ DN mutant, the amplitude of the on relaxation is positive, thus fluctuations that increase the conductance correlate with channel opening ($C \rightarrow O$). Conversely, the amplitude of the off relaxation is negative, thus fluctuations that reduce the conductance correlate with channel closing ($O \rightarrow C$). Mechanistically, a simple interpretation is that the higher conducting conformations of the channel pore raise the energy of the closed state and lower the energy of the open state,

thereby making channel opening faster and channel closing slower. Analogously, the lower conducting conformations do the reverse, raising the energy of the open state while lowering the energy of the closed state, making channel closing faster and channel opening slower. This scenario is depicted in *Figure 9B*.

The impact of fluctuations in the open channel current on the energetics of channel gating can be calculated from the experimentally determined exponential coefficient, $A$, and the magnitude of the open channel fluctuations, $\sigma_i$ according to

$$\Delta\Delta G^* = -kTln\left(1 - \frac{A}{\sigma_i}\right), \tag{6}$$

as derived in Appendix, Part 3. For the $\beta + \delta$ DN mutant, the channel opening transition shows a $\Delta\Delta G^*$ value of 0.84 kT (95% CI = 0.59 kT to 1.18 kT) for the fast relaxation, and a value of 0.82 kT (95% CI = 0.63 kT to 1.08 kT) for the slow relaxation. In contrast, the channel closing transition shows a $\Delta\Delta G^*$ of 0.38 kT (95% CI = 0.22 kT to 0.57 kT) for the fast relaxation and a non-significant value for the slow relaxation of 0.11 kT (95% CI = 0 kT to 0.20 kT). Thus, coupling between open channel current fluctuations and channel gating is asymmetric with respect to the channel opening and closing transitions, with differences in both the sign of the exponential coefficient and the change in the free energy of activation. In sum, our data provide evidence for functional coupling between open channel current and channel gating, and they allow us to quantify the consequences of coupling in terms of changes in the energy barriers for channel gating.

Our studies raise the question of what structures underpin the observed changes in open channel current fluctuations and channel gating. Structural and single channel functional studies on members of the muscle AChR have provided a picture of channel gating in which local agonist-induced structural changes propagate away from the binding site and produce motions in many parts of the protein (*Mukhtasimova et al., 2005*; *Mukhtasimova and Sine, 2007*; *Mukhtasimova et al., 2009*; *Mukhtasimova and Sine, 2018*; *Purohit and Auerbach, 2007a*; *Purohit and Auerbach, 2007b*; *Jha et al., 2007*; *Cadugan and Auerbach, 2010*), ultimately resulting in dilation of the channel pore and permeation of hydrated ions. Recent cryo-electron microscopy studies have further clarified this picture by providing snapshots of the 'closed' and 'open' states of pLGICs, assigned based upon the size of the pore relative to the hydrated ions and molecular dynamics simulations of ion conduction (*Basak et al., 2018*; *Polovinkin et al., 2018*; *Kumar et al., 2020*). Comparing the structures of these closed and open states, one sees that whereas the closed structures show narrow constrictions of the pore at the 9' and −1' positions, the open structures show expansion at these positions. The changes in pore profile between closed and open structures are accompanied by changes in the arrangement of the surrounding transmembrane α-helices. Namely, the four transmembrane α-helices move together as the channel pore expands: an outward blooming of the transmembrane domain. The salt bridge between the pore and peripheral α-helices remains intact throughout the change from the closed to the open state, suggesting the salt bridge helps link the movements of the M2 and M4 α-helices near the cytoplasmic end of the pore, and in so doing, facilitates gating of the channel to ion flow.

The DN mutation is the first example to our knowledge of a residue on the peripheral M4 α-helix that impairs gating of the receptor channel. The DN mutation is further unusual in both diminishing the unitary current amplitude and increasing fluctuations in the current through the open channel. Because the M4 aspartate is located far from the central axis through the pore, we hypothesize that the effects of the DN mutation on ion conduction do not stem from removing the negative charge, but rather from perturbing the pore itself. The two components of the Lorentzian-type fluctuations and the frequency independent constant that are detected in the DN mutants suggest at least three processes contribute to the fluctuations in the open channel current, two of which are also detected in the salt bridge intact receptor. Interestingly the frequency independent component cannot be accounted for by shot noise alone, as the spectral density is roughly fourfold higher than the shot noise predicted from the single channel current in the salt bridge intact receptor, and the component rises further to roughly 30-fold the predicted value in the double DN mutant receptor (shot noise is given by Schottky's formula $S = 2iq$). Thus, the question arises as to what mechanisms underpin the mutant enhanced open channel current fluctuations.

Breaking the salt bridge with DN mutation would be expected to increase the conformational flexibility of the pore lining M2 α-helix, and the increased flexibility could lead to fluctuations in the rate of ion conduction. However, fluctuations arising from increased conformational flexibility of the pore lining α-helix would be expected to be fast, on the order of ~1 ns. Yet experimentally, we observe an increase in open channel fluctuations on slower time scales of ~1 and ~0.1 ms. Hence the mutation may either give rise to or amplify more global and slower structural rearrangements. We consider some potential structural mechanisms that would occur on the experimentally observed time scale below, which may correspond to the structural counterparts of $\sigma_i$.

Within the ion conducting pathway in pLGICs, rings of charged, lumen facing residues select and concentrate ions for transit through the pore. These rings are located in the extracellular, transmembrane, and intracellular domains (*Hansen et al., 2008*; *Imoto et al., 1988*; *Herlitze et al., 1996*). Near the salt bridge there is perhaps the most heavily investigated ring of charged pore-oriented residues—the −1' selectivity filter—and this ring could be involved in the increased open channel current fluctuations observed for the DN mutant. Furthermore, evidence suggests that these residues can change conformation in a manner that affects ion conduction. Specifically, mutations in the −1' position allow the remaining −1' glutamates to adopt different rotameric conformations that produce step changes in ion conductance (*Cymes and Grosman, 2012*). Breaking the salt bridge with the DN mutation could facilitate these rotameric transitions, but discrete steps, if present, are too fast to resolve. Also, the relatively complex network of four glutamates could give rise to both slow and fast current fluctuations, as observed in the open channel power spectrum. Alternatively, the DN mutation may allosterically alter the motion of other rings of charged residues along the ion translocation pathway more distal to the salt bridge.

The increased fluctuations in the open channel current could also arise from fluctuations in the dimensions of the channel lumen. Disrupting the pore-peripheral salt bridge and eliminating the link between the pore-lining M2 and peripheral M4 α-helices could magnify transient changes in the dimensions of the open channel. The size of the channel pore has been shown to correlate with conductance in nicotinic receptors (*Villarroel et al., 1992*; *Kienker et al., 1994*), hence fluctuations in pore dimensions could result in periods of slower and faster ion flow. One could envision these fluctuations occurring near the salt bridge at the −1' position, or perhaps propagating distally to the hydrophobic gate at the 9' position. The two components of Lorentzian-type fluctuations may arise from pore fluctuations at different points along the pore axis.

It is also possible that interactions between permeant ions and the pore may give rise to open channel current fluctuations. While permeant ions can be envisioned to traverse the ion pore uninterrupted once the channel opens, molecular dynamics simulations have shown transient interactions between permeant ions and the pore that momentarily slow the movement of ions (*Wang et al., 2008*). The DN mutation could stabilize these interactions causing transient periods of pore occlusion that manifest as increased fluctuations in the open channel current.

In addition to increasing fluctuations in the open channel current, the DN mutation also alters the kinetics of channel gating, increasing closed lifetimes, decreasing open lifetimes, and reducing the overall channel open probability. A structural basis for this observed change in gating can be inferred from the recently determined high-resolution pLGIC structures. Namely, state specific pLGIC structures show that the salt bridge between the pore and peripheral helices remains intact in both the open and closed states (*Basak et al., 2018*; *Polovinkin et al., 2018*; *Kumar et al., 2020*), suggesting the salt bridge is a structural link that helps couple movements of the M2 and M4 α-helices near the cytoplasmic end of the transmembrane domain during gating. Thus, breaking the salt bridge decouples the M2 from the M4 α-helix in the region near the cytoplasmic end of the pore such that M2 less readily expands to allow ion flow and more readily collapses to occlude flow.

Unlike the novel effect of the DN mutation on ionic conductance, there are many examples of mutations in the transmembrane domains that alter channel gating kinetics. Placing the DN mutation in the context of site-directed and naturally occurring pathogenic mutations, we see there are multiple mutations in the M2 and M4 domains that alter channel gating (*Herlitze et al., 1996*; *Villarroel et al., 1992*; *Kienker et al., 1994*). In contrast to the DN mutant, however, most of these 'gating' mutations are located further toward the extracellular end of the transmembrane domain and enhance rather than suppress channel gating efficiency (*Lasalde et al., 1996*; *Labarca et al., 1995*; *Engel and Sine, 2005*). In further contrast, these gain-of-function mutations target hydrophobic or weakly polar residues and appear to alter inter-helical motions that accompany channel

gating. Less work has been done in the region of the salt bridge near the cytoplasmic end of the channel pore, with the notable exception of the −1' glutamates in the context of ion selectivity (*Cymes and Grosman, 2012*). Interestingly, in closer proximity to the salt bridge studied here, within the linker joining the M1 and M2 α-helices, both site-directed and naturally occurring pathogenic mutations have been identified that alter channel gating (*Lynch et al., 1997*; *Saul et al., 1999*; *Shen et al., 2020*). Like the DN mutation, these mutations diminish channel gating efficiency, but to a lesser extent than the DN mutation. Thus, as observed for the DN mutation in the M4 domain, mutations in other transmembrane domains impact the kinetics of channel gating.

In summary, we show that two fundamental facets of muscle AChR function, current through the open channel and gating of the channel, are coupled to a common structural element: a conserved pore-peripheral salt bridge. We further show that this structural coupling is accompanied by functional coupling between fluctuations in ion flow and transitions underpinning channel gating. We find that functional coupling is stronger for the channel opening transition compared to the closing transition, and quantify this in terms of the changes in the energy barriers for channel gating. A model is presented in which disrupting the salt bridge magnifies an intramolecular motion that is simultaneously coupled to changes in open channel current fluctuations and the energy barriers for channel gating. Thus, the pore-peripheral salt bridge emerges as a critical linkage governing ion flow, channel gating, and their coupling.

## Materials and methods

### Expression and mutagenesis of the adult human muscle receptor

Bosc-23 cells (*Pear et al., 1993*), a cell line derived from human embryonic kidney (HEK)293 cells, were used to express wild-type and mutant muscle receptor subunits. Cells were maintained at 37°C in Dulbecco's modified Eagle's medium with 10% fetal bovine serum. When cells reached ~50% confluency, they were transfected with cDNAs encoding wild-type or mutant muscle receptor subunits that were installed within cytomegalovirus expression vector pRBG4 (*Lee et al., 1991*). The subunit cDNAs were transfected in a 2:1:1:1 α:β:δ:ε ratio using calcium phosphate precipitation. Additionally, cells were also transfected with a mammalian expression vector containing a cDNA encoding GFP to identify recipient cells for patch-clamp recordings. Mutations were generated using the QuickChange site-directed mutagenesis kit (Agilent) and were confirmed by sequencing. Patch clamp recordings were made 24–72 hr following transfection.

### Single channel recordings

Single channel currents were recorded in the cell-attached patch configuration with a membrane potential of −70 mV for analysis of open and closed dwell times and −120 mV for analysis of open channel current fluctuations. Patch pipettes were fabricated from type 8250 glass (King Precision Glass), coated with Sylgard 184 (Dow Corning), and heat polished to yield resistances of 5–8 megaohms. Extracellular solutions contained (mM) 142 KCl, 5.4 NaCl, 1.7 $MgCl_2$, 1.8 $CaCl_2$, and 10 HEPES, adjusted to pH 7.4 with NaOH. For recordings used for kinetic analysis, pipettes were filled with the same solution without $CaCl_2$. For recordings of open channel current fluctuations, pipettes were filled with (mM) 80 KF, 20 KCl, 40 K-aspartate, 2 $MgCl_2$, 1 EGTA, and 10 HEPES, adjusted to pH 7.4 with KOH. Concentrated stock solutions of ACh were stored at −80°C until diluted for use on the day of each experiment.

Single channel currents were recorded using an Axopatch 200B patch clamp amplifier with the gain set to 100 mv/pA and the internal Bessel filter at 10 kHz. Continuous stretches of channel openings were recorded at a sample interval of 2 µs using a National Instruments model BNC-2090 A/D converter with a PCI6111e acquisition card and recorded onto the hard drive of a PC computer using the program Acquire (Bruxton Corporation).

### Kinetic analysis

Channel openings and closings were detected with a half-amplitude threshold criterion using the program TAC4.2.0 (Bruxton Corporation), as described previously (*Mukhtasimova et al., 2016*). Dwell time histograms are displayed with a logarithmic horizontal axis and a square root vertical axis for better visualization of distinct exponential components (*Sigworth and Sine, 1987*).

Representative histograms from a single patch are shown. Mean weights and time constants from three independent patches are provided in *Supplementary file 1b-h*. Critical closed times were used to define clusters of openings from a single receptor channel free of closed dwell times due to desensitization, and were set from the intersection of the longest closed component with the adjacent briefer component. Open probability was then computed as the fraction of time spent in the open state within a cluster of channel openings. For each concentration of ACh, three independent patches were analyzed and the open probability of all clusters was averaged. Open probability versus ACh concentration was fit by a simple sequential bind-bind-gate model with response function $\frac{\theta}{1+\theta+\frac{2K}{[ACh]}+\frac{K^2}{[ACh]^2}}$ using GraphPad Prism version 8.

## Open channel fluctuations analysis

### Data Segmentation

To quantify the magnitude and frequency content of open channel current fluctuations, we emulated the analysis done by Sigworth in his first paper on open channel noise (*Sigworth, 1985*). Briefly, at least fifty 40 ms segments of open channel current and an equivalent segment of flanking baseline were extracted from continuous recordings and imported from TAC 4.2.0 into MATLAB 2019b. Care was taken to segment stretches of openings that had minimal brief closings within the open channel current. However, eliminating these closings entirely was impossible; therefore, unresolved closing events within stretches of open channel current were identified as points that fell below half the amplitude of the open channel current. These points and two flanking points on either side were removed from the open channel segments before further analysis.

### Magnitude analysis

To obtain a measure of the magnitude of the open channel current fluctuations, the variance of the open channel current segments and flanking baseline current segments were calculated. For each opening, the baseline value was subtracted from the open channel value. The resultant difference is therefore independent of changes in the background noise of a given recording. From the individual difference variances, the standard deviation was computed and an average for all openings was taken, yielding the average open channel standard deviation as reported in *Figure 6c*.

### Frequency analysis

A Fast Fourier Transform was performed on each 40 ms open channel and baseline segment and squared to obtain a power spectrum. Baseline power spectra were subtracted from open channel spectra to eliminate frequency content from the baseline recorded noise. Analogous to the standard deviation computations, this was done for open channel and baseline segments and then averaged to achieve an average difference spectrum. To calibrate the spectrum for the frequency response of our recording system, a series combination of capacitor (0.01 µF) and resistor (20 MΩ) was connected between the head stage of the patch clamp and the reference voltage terminal. Power spectra calculated from this reference input yielded the expected spectrum with a plateau value of 4kT/R until rolling off at high frequencies (*Figure 6—figure supplement 1*). The resultant reference spectrum was then normalized to the 4kT/R value. We then divided point-by-point the experimental difference spectrum by the normalized reference spectrum to obtain the final calibrated spectrum. Lorentzian fits to the final calibrated spectra were optimized within MATLAB using the non-linear least squares optimization.

## Coupling analysis

For mutant and wild-type receptors, opening and closing transitions were segmented in 8 ms portions from TAC 4.2.0 and imported into MATLAB. Transitions were aligned at the point where the current reached half amplitude and averaged to obtain a mean channel opening and mean channel closing profile as had been done previously (*Sigworth, 1986*). The mean channel opening and mean channel closing profiles were aligned at the half amplitude point and summed. In the absence of coupling, the sum is expected to appear as fluctuations superimposed upon a horizontal line. In the presence of coupling, the sum is expected to show exponential relaxations upon channel opening and upon channel closing, with time constants that relate to the Lorentzian cut off frequencies ($f_c$) of

any correlated noise according to the relationship $1 + Ae^{-2\pi f_c t}$, where $A$ is a coefficient representing the degree of coupling between the open channel current fluctuations and the rate constant for the channel gating step, either opening or closing. A global fit of power spectra and opening and closing relaxations was performed in MATLAB using non-linear least squares optimization.

## Acknowledgements

This research was supported by National Institutes of Health grant NS-031744 to S.M. Sine and NS-115358 to JR Strikwerda.

## Additional information

### Funding

| Funder | Grant reference number | Author |
|---|---|---|
| National Institute of Neurological Disorders and Stroke | NS-031744 | John R Strikwerda |
| National Institute of Neurological Disorders and Stroke | NS-115358 | Steven M Sine |

The funders had no role in study design, data collection and interpretation, or the decision to submit the work for publication.

### Author contributions

John R Strikwerda, Conceptualization, Formal analysis, Funding acquisition, Investigation, Writing - original draft, Writing - review and editing; Steven M Sine, Conceptualization, Supervision, Funding acquisition, Writing - original draft, Writing - review and editing

### Author ORCIDs

John R Strikwerda (iD) https://orcid.org/0000-0001-6155-7919
Steven M Sine (iD) https://orcid.org/0000-0002-0594-6052

### Decision letter and Author response

Decision letter https://doi.org/10.7554/eLife.66225.sa1
Author response https://doi.org/10.7554/eLife.66225.sa2

## Additional files

### Supplementary files

• Supplementary file 1. Fitted binding and gating parameters and event lifetime summaries for wild-type and DN receptors. (A) Binding and gating equilibrium constants for the wild-type and salt bridge mutated (DN) receptors. Constants determined by fitting of the open probability data to a sequential bind-bind-gate mechanism. Open probability data and fits shown in *Figure 2* in main text. (B) Binding and gating equilibrium constants for β-DN + δ-DN and charge exchanged DKKD receptors. Constants determined by fitting of the open probability data to a sequential bind-bind-gate mechanism. Open probability data and fits shown in *Figures 3* and *4* in main text. (C) Average dwell time component weights (W) and time constants (T) of the wild-type muscle receptor at 3, 10, 30, 100, 300 µM [ACh]. Values are averages of three independent recordings with the corresponding standard deviation (SD). (D) Average dwell time component weights (W) and time constants (T) of the β-DN receptor at 3, 10, 30, 100, 300 µM [ACh]. Values are averages of three independent recordings with the corresponding standard deviation (SD). (E) Average dwell time component weights (W) and time constants (T) of the δ-DN receptor at 3, 10, 30, 100, 300 µM [ACh]. Values are averages of three independent recordings with the corresponding standard deviation (SD). (F) Average dwell time component weights (W) and time constants (T) of the beta- + delta- DN receptor at 3, 10, 30, 100, 300 µM [ACh]. Values are averages of three independent recordings with the corresponding standard deviation (SD). (G) Average dwell time component weights (W) and time

constants (T) for the δ-DKKD receptor at 3, 10, 30, 100, 300 μM [ACh]. Values are averages of three independent recordings with the corresponding standard deviation (SD). (H) Average dwell time component weights (W) and time constants (T) of the β-DKKD receptor at 3, 10, 30, 100, 300 μM [ACh]. Values are averages of three independent recordings with the corresponding standard deviation (SD).

- Supplementary file 2. Power spectra and gating relaxation fits for salt bridge intact and DN receptors. (A) Frequency and magnitude of open channel current fluctuations in wild type and DN receptors. $S_s$, $S_f$, $f_{cs}$, $f_{cf}$, $S_c$ determined from least squares fit of *Equation 1* in main text to open channel current power spectra. SD/A indicates open channel current standard deviation relative to channel amplitude. Spread is shown as 95% confidence or standard deviation as indicated in table. Fits obtained from the average spectrum of at least 50 open channel segments with baseline spectra subtracted. (B) Power spectra and open channel current relaxations of the β-DN + δ-DN receptor. Best fit parameters obtained from simultaneous least squares fit of *Equation 1* from main text, *Equations A7 and A8* to the power spectrum and open channel current relaxations for the β-DN + δ-DN receptor.

- Transparent reporting form

## Data availability

All data relevant to this work is presented in the manuscript and supporting files.

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

## Appendix 1

## Part 1: Predicted relaxations around gating transitions

To describe the consequences of coupling between open channel current fluctuations and channel gating, we present an expression for the relaxation of the average unitary current upon transition between closed and open current levels. That is, we define the expected value of the open channel current near a channel opening or closing transition, $E[i(t_1)|closed\ at\ t_0]$ or $E[i(t_1)|open\ at\ t_0]$. This problem was considered previously in *Sigworth, 1986*, and we refer the reader there for a detailed theoretical discussion on the topic. Here, we present the result of the derivation in *Sigworth, 1986* and introduce the notation used throughout this work.

First, the open channel current, $i$, as well as the channel closing and opening rate constants $\alpha$ and $\beta$ are assumed to be stochastic processes that fluctuate about their means according to

$$i(t) = i_0 \left[ 1 + k_{i\boldsymbol{x}_s} \boldsymbol{x}_s(t) + k_{i\boldsymbol{x}_f} \boldsymbol{x}_f(t) \right] \tag{EA1}$$

$$\alpha(t) = \alpha_0 \left[ 1 + k_{\alpha\boldsymbol{x}_s} \boldsymbol{x}_s(t) + k_{\alpha\boldsymbol{x}_f} \boldsymbol{x}_f(t) \right] \tag{EA2}$$

$$\beta(t) = \beta_0 \left[ 1 + k_{\beta\boldsymbol{x}_s} \boldsymbol{x}_s(t) + k_{\beta\boldsymbol{x}_f} \boldsymbol{x}_f(t) \right], \tag{EA3}$$

where $\boldsymbol{x}_s(t)$ and $\boldsymbol{x}_f(t)$ are zero mean stationary random processes representing the intramolecular motions that underlie the slow and fast open channel current fluctuations observed in power spectra. And the k's represent coupling coefficients between the motions, $\boldsymbol{x}_s(t)$ and $\boldsymbol{x}_f(t)$, and either the open channel current, $i$, or the channel gating rate constants, $\alpha$ and $\beta$. From *Equations 2 and 3* we can define the relative rms values of $i$, $\alpha$, and $\beta$ resulting from motion $\boldsymbol{x}$ as

$$\sigma_i = k_{ix}\sigma_x \tag{EA4}$$

$$\sigma_\alpha = k_{\alpha x}\sigma_x \tag{EA5}$$

$$\sigma_\beta = k_{\beta x}\sigma_x. \tag{EA6}$$

Given *Equation 1* of the main text and S1-6, the expected value of the open channel current near the gating transition, as shown in *Sigworth, 1986*, is an exponential function, with time constants that correspond to observed Lorentzian cut-off frequencies multiplied by $2\pi$, and amplitudes that are the product of the rms value of $i$ and the rate constants $\alpha$ or $\beta$. As a result, we can express the expectation value of $i$ near the closing transition as the following two component exponential function.

$$E[i(t_1)|closed\ at\ t_0] = i_0 \left[ 1 + A_{\alpha s} e^{-|2\pi f_{cs} t|} + A_{\alpha f} e^{-|2\pi f_{cf} t|} \right] \tag{EA7}$$

And similarly for the channel opening transition,

$$E[i(t_1)|open\ at\ t_0] = i_0 \left[ 1 + A_{\beta s} e^{-|2\pi f_{cs} t|} + A_{\beta f} e^{-|2\pi f_{cf} t|} \right], \tag{EA8}$$

where

$$A_{\alpha s} = \sigma_{\boldsymbol{x}_s}^2 k_{\alpha\boldsymbol{x}_s} k_{i\boldsymbol{x}_s} \tag{EA9}$$

$$A_{\alpha f} = \sigma_{\boldsymbol{x}_f}^2 k_{\alpha\boldsymbol{x}_f} k_{i\boldsymbol{x}_f} \tag{EA10}$$

$$A_{\beta s} = \sigma_{\boldsymbol{x}_s}^2 k_{\beta\boldsymbol{x}_s} k_{i\boldsymbol{x}_s} \tag{EA11}$$

$$A_{\beta f} = \sigma_{\boldsymbol{x}_f}^2 k_{\beta\boldsymbol{x}_f} k_{i\boldsymbol{x}_f}. \tag{EA12}$$

Note that the amplitude of the relaxation, $A$, depends upon the fluctuations in the molecular motions $x$ ($\sigma_x$), the degree to which these fluctuations are coupled to changes in the open channel current ($k_{ix}$) and the opening and closing rate constants ($k_{\alpha x}$ or $k_{\beta x}$). If there is no coupling between motion $x$ and either the open channel current or the rate constants governing channel gating, the amplitudes reduce to zero. Thus, only a motion that impacts both the rate of ion flow through the receptor channel and the rate constants for channel gating will show relaxations in the average unitary current.

The experimentally determined average unitary current for the DN mutant can be fit by *Equations A7 and A8*, yielding amplitudes with a positive sign for the opening transition and a negative sign for the closing transition (see *Figure 8*, main text). By definition, the coupling coefficients between motion $x$ and the open channel current, $k_{ix}$, are positive. Therefore, the coupling coefficient between motion $x$ and the opening rate constant, $k_{\beta x}$, is also positive. In contrast, the coupling coefficient between motion $x$ and the closing rate constant, $k_{\alpha x}$, is negative. Hence, motion $x$ has opposing effects on channel opening compared to channel closing. This can be readily understood in terms of the energetics of gating depicted in *Figure 9*. Conformations associated with increased conductance (green) raise the energy of the closed state and lower energy of the open state, making channel opening faster and channel closing slower. The opposite is observed for conformations associated with reduced conductance (red).

## Part 2: Mechanistic interpretation of the on and off relaxations

Relaxations in the average unitary current are readily detected in the DN mutant receptor, but not in the wild type receptor. Thus, the question arises of what underpins the greater relaxations in the mutant compared to the wild type. Within the mechanism described in Part 1, the mutation could potentially increase one or more of the following three parameters: the fluctuations of motion $x$ ($\sigma_x$), the coupling of $x$ to open channel current ($k_{ix}$), or the coupling of $x$ to channel gating ($k_{\alpha x}$ or $k_{\beta x}$). Indeed, each of these scenarios would increase the amplitude of the relaxations in the average unitary current, as observed experimentally. However, of these three scenarios we can exclude one of them. This is because a change in the coupling of $x$ to channel gating cannot explain the increase in the magnitude of open channel current fluctuations observed experimentally. In other words, only a mutation-induced change in either $\sigma_x$ or $k_{ix}$ can explain both the increased amplitude of the relaxations in the average unitary current and the increase in the magnitude of the open channel current fluctuations. Therefore, we consider two scenarios, one where the mutation alters $\sigma_x$ and another where it alters $k_{ix}$. For each scenario, we calculate the expected relaxation amplitudes for the wild-type receptor and compare the predicted relaxations to the corresponding experimental average unitary current.

For the case in which the DN mutation increases $\sigma_x$, while not affecting $k_{ix}$, we can write

$$k_{ixwt} = k_{ixDN}.$$

Additionally, coupling of motion $x$ to channel gating is assumed to remain unchanged between mutant and wild-type receptors so that for the channel closing transition

$$k_{\alpha xwt} = k_{\alpha xDN}.$$

Using these relationships, the exponential coefficient for the relaxation of the wild type receptor can be expressed as

$$A_{wt} = k_{\alpha x_{wt}} \sigma_{x_{wt}} k_{ix_{wt}} \sigma_{x_{wt}} = k_{\alpha x_{DN}} \sigma_{x_{wt}} k_{ix_{DN}} \sigma_{x_{wt}}.$$

Here,

$$\sigma_{i_{wt}} = k_{ix_{wt}} \sigma_{x_{wt}} = k_{ix_{DN}} \sigma_{x_{WT}}.$$

Therefore,

$$\sigma_{x_{wt}} = \frac{\sigma_{i_{wt}}}{k_{ix_{DN}}}.$$

Combining these expressions yields

$$A_{wt} = k_{\alpha x_{DN}} \sigma_{x_{wt}} \sigma_{i_{wt}} = \frac{k_{\alpha x_m}}{k_{ix_m}} \sigma_{i_{wt}}^2.$$

Combining *Equations A4 and A9*, yields an expression for the ratio of coupling coefficients,

$$\frac{k_{\alpha x_{DN}}}{k_{ix_{DN}}} = \frac{A_{DN}}{\sigma_{i_{DN}}^2}.$$

Therefore,

$$A_{wt} = \left(\frac{\sigma_{i_{wt}}}{\sigma_{i_{DN}}}\right)^2 A_{DN}.$$

Thus, the wild type relaxation amplitudes can be predicted from experimentally measurable quantities. For the case in which only $\sigma_x$ changes, the wild type amplitudes are scaled by a factor of $\left(\frac{\sigma_{iwt}}{\sigma_{iDN}}\right)^2$ relative to the amplitudes for the DN mutant, and are predicted to be ~6 % the size. Relaxations calculated according to the predicted amplitudes and the experimentally observed cutoff frequencies, shown in *Figure 8—figure supplement 3*, fall within the noise envelope of the experimental data.

Next we consider the case in which the DN mutation changes $k_{ix}$ but does not affect $\sigma_x$. Therefore,

$$\sigma_{x_{wt}} = \sigma_{x_{DN}}.$$

Again, we assume the mutation does not alter coupling of motion **x** to the rate constants for channel gating so that for the channel closing transition

$$k_{\alpha x_{wt}} = k_{\alpha x_{DN}}.$$

Now the expression for the relaxation amplitude for the wild type receptor becomes

$$A_{wt} = k_{\alpha x_{wt}} \sigma_{x_{wt}} k_{ix_{wt}} \sigma_{x_{wt}} = k_{\alpha x_{DN}} \sigma_{x_{DN}} k_{ix_{wt}} \sigma_{x_{DN}}.$$

Recalling that

$$\sigma_{i_{wt}} = k_{ix_{wt}} \sigma_{x_{wt}} = k_{ix_{wt}} \sigma_{x_{DN}}$$

$$\sigma_{\alpha_{DN}} = k_{\alpha x_{DN}} \sigma_{x_{DN}}.$$

Therefore,

$$A_{wt} = \sigma_{\alpha_{DN}} \sigma_{i_{wt}}.$$

The quantity $\sigma_\alpha DN$ can be determined from the relaxation amplitudes for the DN mutant according to

$$\sigma_{\alpha_{DN}} = \frac{A_{DN}}{\sigma_{i_{DN}}}.$$

Therefore,

$$A_{wt} = \frac{\sigma_{i_{wt}}}{\sigma_{i_{DN}}} A_{DN}.$$

Thus, the wild type relaxation amplitudes can be predicted from experimentally measurable quantities. For the case in which only $k_{ix}$ changes, the wild type amplitudes are scaled by a factor of $\frac{\sigma_{iwt}}{\sigma_{iDN}}$ relative to the amplitudes for the DN mutant, and are predicted to be ~25 % the size. Relaxations calculated according to the predicted amplitudes and the experimentally observed cutoff frequencies, shown in *Figure 8—figure supplement 3*, are again similar to the noise envelope of the experimental data. Thus, our observations for the wild type receptor are consistent with the mutation changing either motion **x** or its coupling to the open channel current.

Finally, we consider a scenario in which the mutation impacts both $\sigma_x$ and $k_{ix}$. Recall the expressions for the relaxation amplitude and rms open channel current that are entirely in terms of motion **x**,

$$A = k_{\alpha x}k_{ix}\sigma_x^2$$

$$\sigma_i = k_{ix}\sigma_x.$$

Thus $A$ is proportional to $k_{ix}$ and the square of $\sigma_x$. Meanwhile $\sigma_i$ is directly proportional to both $\sigma_x$ and $k_{ix}$. As a result, we can see that a change in either $\sigma_x$ or $k_{ix}$ will have equivalent impact on $\sigma_i$, but a change in $\sigma_x$ will alter $A$ to a greater extent than a change in $k_{ix}$. Hence, for the observed decrease in $\sigma_i$ between mutant and wild type, the maximum change in $A$, and therefore the minimum value of $A_{wt}$, will occur when only $\sigma_x$ changes. Conversely for the observed decrease in $\sigma_i$ between mutant and wild type, the minimum change in $A$, and therefore the maximum value of $A_{wt}$, will occur when there is no change in $\sigma_x$. As we demonstrated above, these scenarios represent situations in which $A_{wt}$ is scaled by either $\left(\frac{\sigma_i wt}{\sigma_i DN}\right)^2$ or $\left(\frac{\sigma_i wt}{\sigma_i DN}\right)$, respectively. Therefore, a scenario where the DN mutation impacts both $\sigma_x$ and $k_{ix}$ represents a scenario where the amplitude is scaled by a factor between $\left(\frac{\sigma_i wt}{\sigma_i DN}\right)^2$ and $\left(\frac{\sigma_i wt}{\sigma_i DN}\right)$.

## Part 3: Calculating changes in the activation energy for channel gating

In this section we relate the experimentally determined relaxation amplitudes ($A$) to changes in the free energy of activation for the gating transitions. We derive changes in free energy for the case in which the fast fluctuating process is coupled to the opening transition, and then apply the result to coupling of the slow and fast fluctuating processes to the opening and closing transitions.

To begin, the amplitude of the relaxation, $A$, can be written as the product of the rms amplitude of the Lorentzian fluctuations and the relative standard deviation of rate constant (**Sigworth, 1986**),

$$A_{\beta f} = \sigma_{if}\sigma_{\beta f}. \tag{EA13}$$

Rearranging and solving for the standard deviation of the rate constant,

$$\sigma_{\beta f} = \frac{A_{\beta f}}{\sigma_{if}}.$$

The rms current, $\sigma_{if}$, while not explicitly measured can be solved according to

$$\sigma_{if} = \sqrt{\frac{\pi S_f f_{cf}}{2i_0^2}}. \tag{EA14}$$

Where $f_{cf}$ is the fast Lorentzian cut off frequency, $S_f$ is the magnitude of the fast Lorentzian component, and $i_0$ is the single channel current amplitude. Combining **Equations A13 and A14**, we obtain

$$\sigma_{\beta f} = \frac{A_{\beta f}}{\sqrt{\frac{\pi S_f f_{cf}}{2i_0^2}}}.$$

Note that all the constants—$A_{\beta f}, S_f, f_{cf}, i_0$—have been determined experimentally. The relative standard deviation of the rate constant, $\sigma_{\beta f}$, can then be used to calculate $G$ for the gating transition.

$$\Delta\Delta G^* = \Delta G_f^* - \Delta G_0^*$$

$$= -kTln\left(\frac{(1 - \sigma_{\beta f})*\beta}{AC}\right) + kTln\left(\frac{\beta}{AC}\right)$$

$$= -kTln\left(1 - \sigma_{\beta f}\right) - kTln\left(\frac{\beta}{AC}\right) + kTln\left(\frac{\beta}{AC}\right)$$

$$\Delta\Delta G^* = -kTln\left(1 - \sigma_{\beta f}\right),$$

where $\Delta G_0^*$ represents the mean activation energy, $\Delta G_f^*$ represents the heightened energy barrier during conformational fluctuations associated with the lower bound of the rms variations in the rate constant, and $AC$ is the Arrhenius constant. Below we calculate $\Delta\Delta G^*$ for the slow and fast fluctuating processes for the opening and closing transitions.

$$\sigma_{\beta f} = \frac{A_{\beta f}}{\sqrt{\frac{\pi S_f f_{cf}}{2i_0^2}}}$$

$$\sigma_{\beta f} = \frac{0.073}{\sqrt{\frac{\pi * 4.25 * 10^{-28}\left(\frac{A^2}{Hz}\right) * 2.85 * 10^3 (Hz)}{2 * (1.08 * 10^{-11} (A))^2}}}$$

$$\sigma_{\beta f} = 0.57$$

$$\Delta\Delta G^* = -kTln\left(1 - \sigma_{\beta f}\right)$$

$$\Delta\Delta G^* = -kTln(0.43) = 0.84 \text{kT}$$

$$95\% \, CI = 0.59 kT \text{ to } 1.18 kT$$

$$\sigma_{\beta s} = \frac{A_{\beta s}}{\sqrt{\frac{\pi S_s f_{cs}}{2i_0^2}}}$$

$$\sigma_{\beta s} = 0.560$$

$$\Delta\Delta G^* = -kTln\left(1 - \sigma_{\beta s}\right)$$

$$\Delta\Delta G^* = -kTln(0.5044) = 0.7082 \text{kT}$$

$$95\% \, CI = 0.340.63 kT \text{ to } 1.08 kT$$

$$\sigma_{\alpha f} = \frac{A_{\alpha f}}{\sqrt{\frac{\pi S_f f_{cf}}{2i_0^2}}}$$

$$\sigma_{\alpha f} = \frac{0.0405}{\sqrt{\frac{\pi * 5.464.25 * 10^{-28}\left(\frac{A^2}{Hz}\right) * 3.002.85 * 10^3 (Hz)}{2 * (1.078 * 10^{-11} (A))^2}}}$$

$$\sigma_{\alpha f} = 0.310$$

$$\Delta\Delta G^* = -kTln\left(1 - \sigma_{\alpha\beta f}\right)$$

$$\Delta\Delta G^* = -kTln(0.7069) = 0.368 \text{kT}$$

$$95\% \, CI = 0.1922kT \, \text{to} \, 0.57kT$$

$$\sigma_{\alpha s} = \frac{A_{\alpha s}}{\sqrt{\frac{\pi S_s f_{cs}}{2i_0^2}}}$$

$$\sigma_{\alpha s} = \frac{0.00375}{\sqrt{\frac{\pi * 8.141.55 * 10^{-287} \left(\frac{A^2}{Hz}\right) * 17714(Hz)}{2*(1.07*10^{-11}(A))^2}}}$$

$$\sigma_{\alpha s} = 0.1108$$

$$\Delta\Delta G^* = -kTln\left(1 - \sigma_{\alpha\beta s}\right)$$

$$\Delta\Delta G^* = -kTln(0.9289) = 0.0911\text{kT}$$

$$95\% \, CI = 0.0018kT \, \text{to} \, 0.20kT$$

Thus, fluctuations in motion **x** alter the energy landscape for channel gating asymmetrically. The $\Delta\Delta G^*$ values are near 0.8kT for the closed to open transition, but the maximum value for the open to closed transition is 0.38 kT. Furthermore, while the contribution of the slow and fast current fluctuations are roughly equal in terms of their impact on the energy of the channel opening transition, the slow fluctuations have a lesser impact on the channel closing transition.

