## [Decision Letter]

**Acceptance summary:**

The authors provide evidence that fluctuations in ionic current through the open pore of a mutant acetylcholine-receptor ion channel are correlated with current transients during pore opening/closing conformational changes. The data are of good quality, and their rigorous analysis suggests that pore gating is coupled to fluctuations of ion conduction. Although the observations must be extrapolated to wild-type channels, they will be of fundamental interest to the ion channel community.

**Decision letter after peer review:**

Thank you for submitting your article "Unmasking coupling between channel gating and ion permeation in the muscle nicotinic receptor" for consideration by *eLife*. Your article has been reviewed by three peer reviewers, including Marcel P Goldschen-Ohm as the Reviewing Editor and Reviewer #1, and the evaluation has been overseen by Kenton Swartz as the Senior Editor. The following individuals involved in review of your submission have agreed to reveal their identity: Fred Sigworth (Reviewer #2); Angelo Keramidas (Reviewer #3).

Essential Revisions:

1) Main problem: You need to correct for the effect of the Bessel filter, whose rolloff looks a lot like a Lorentzian with cutoff near its cutoff frequency. You could use a complicated measurement of the system frequency response to correct the power spectra. Alternatively, here is a very simple method: connect the series combination of a resistor and capacitor between the headstage input terminal and the Vref (not ground) terminal. Good values are R = 20 to 50 megohms and C=.01 uF. The result is that you should measure the flat spectrum of resistor noise, spectral density = 4kT/R from ~1 Hz to high frequencies. You can then compensate for high-frequency rolloff by dividing your channel noise spectrum by the rolloff of this one, which should be obvious around 10kHz. The reviewers are happy to be contacted by the authors about how to do the spectrum calibration.

Further suspicion for an artifact is because there must be a flat spectral component extending up to the megahertz range due to shot and thermal noise in the channel. Even if your second Lorentzian disappears, the high-frequency asymptote of the spectrum might be interesting to compare.

2) Figure 8 – correlation of the low frequency component with the ON and OFF transients is visually somewhat ambiguous. Please show fits with the high frequency component only for comparison. This is doubly important if the high frequency component contains any artifact from the filter as discussed above.

3) The epsilon subunit mutation (Introduction). Where is it in the channel structure? Given its large effect on conductance and open time, why don't you think it will change the relevant behavior of the channel? It would be helpful for most readers if the authors could comment on the mechanism for this mutation in the text and briefly discuss what impact if any it might have on interpretation of the data with regards to WT channels.

4) Throughout: "wild-type" is misleading, as it should be epsilon-T264P.

5) Introduction. Please include sequence alignments of a few receptors in Figure 1 to show conservation of the salt-bridge across family and generality of the study. Also, please explicitly identify the residue position number of the acidic residue on M4 in the text.

6) Statements such as "These results localize the primary effect of the salt bridge disrupting mutation…" are model dependent, and the issue is with the word "primary". Which model do the authors have in mind? In the usual sequential model the open-closed behavior at saturating agonist is indeed due only to the final opening step; but is the change sufficient to explain (at least approximately) the shifted dose-response curve? Or is a substantial change in binding constants required as well? Similarly, the charge-reversal mutation DKKD is presented as nearly equivalent to the wildtype channel, but it is in fact a partial rescue, and the completeness of its rescue might best be evaluated with a model.

7) A related issue is the existence of a new, longer open time component at high agonist concentrations that is mentioned in the Results. Please show some data to give the reader an idea of its importance. Did this affect the saturating open probability calculations?

8) Fluctuations that are so slow (~1 ms) are unlikely to be arise from the breaking of a salt bridge that (one would expect) would leave a helix flopping in the breeze on a ~1 ns timescale. So there's something complicated going on here with slow rearrangements of the protein that have only small effects on the conductance. A discussion of the relation of these timescales to potential mechanisms would be helpful for many readers.

---

## [Author Response]

Essential revisions:1) Main problem: You need to correct for the effect of the Bessel filter, whose rolloff looks a lot like a Lorentzian with cutoff near its cutoff frequency. You could use a complicated measurement of the system frequency response to correct the power spectra. Alternatively, here is a very simple method: connect the series combination of a resistor and capacitor between the headstage input terminal and the Vref (not ground) terminal. Good values are R = 20 to 50 megohms and C=.01 uF. The result is that you should measure the flat spectrum of resistor noise, spectral density = 4kT/R from ~1 Hz to high frequencies. You can then compensate for high-frequency rolloff by dividing your channel noise spectrum by the rolloff of this one, which should be obvious around 10kHz. The reviewers are happy to be contacted by the authors about how to do the spectrum calibration.Further suspicion for an artifact is because there must be a flat spectral component extending up to the megahertz range due to shot and thermal noise in the channel. Even if your second Lorentzian disappears, the high-frequency asymptote of the spectrum might be interesting to compare.

We are deeply grateful to the reviewers for pointing out the need to calibrate our power spectra, and for providing instruction on the method to do this. As suggested, a series combination of capacitor (0.01 μF) and resistor (20 MΩ) was connected between the patch clamp head stage and the reference voltage terminal. Power spectra were calculated from this reference input yielding the expected flat spectrum with a plateau spectral density of 4kT/R until rolling off at high frequencies. See Figure 6—figure supplement 1. The resultant spectrum was then normalized to the 4kT/R value. We then divided point-by-point the experimental difference spectra by this reference spectrum to obtain the calibrated spectra.

Additionally, in the process of performing these calibrations, it became apparent that following the 10 kHz Bessel filter on the patch clamp, we were unnecessarily filtering the digitized data with a 10 kHz Gaussian digital filter. Therefore, we reanalyzed the data omitting the additional 10 kHz digital filtering to retain as much high frequency signal as possible. In so doing we obtained moderately better fits to the high frequency portions of both the power spectra and current relaxation profiles.

After performing the calibration, the spectra from the DN mutant receptors are still best described by two Lorentzian components, however with an additional frequency independent component. The magnitude of this component does indeed increase with the DN mutation. In contrast, the spectrum of the salt bridge intact receptor, is now best fit by a single Lorentzian component plus a frequency independent constant, as described in the text.

2) Figure 8 – correlation of the low frequency component with the ON and OFF transients is visually somewhat ambiguous. Please show fits with the high frequency component only for comparison. This is doubly important if the high frequency component contains any artifact from the filter as discussed above.

Agree. In Figure 8—figure supplement 1, we show fits with just the low frequency component and high frequency component. Neither component alone provides a satisfactory fit to the on and off relaxations.

3) The epsilon subunit mutation (Introduction). Where is it in the channel structure? Given its large effect on conductance and open time, why don't you think it will change the relevant behavior of the channel? It would be helpful for most readers if the authors could comment on the mechanism for this mutation in the text and briefly discuss what impact if any it might have on interpretation of the data with regards to WT channels.

Further discussion regarding the location and mechanism of the epsilon T264P background mutation has been added to the text. Briefly, the epsilon T264P mutation is located at the 12’ position of M2, which is well above the 0’ position of the basic residue that forms the salt bridge. Additionally, the arrangement of the muscle AChR subunits is such that the T264P harboring epsilon subunit does not directly contact either the β or δ subunits that contain the DN mutation. The epsilon T264P mutation has been shown to have no effect on single channel conductance (Cymes and Grosman, 2011). The increase in single channel amplitudes between Figures 5 and 6 arises from an increased membrane potential and differences in the ionic composition of the pipette solution, as described in the Materials and methods. The reason for these changes was to increase the signal to noise ratio of open channel fluctuations.

4) Throughout: "wild-type" is misleading, as it should be epsilon-T264P.

Agree. We now refer to constructs with the epsilon T264P mutation, with or without the DN mutation, as either “salt bridge intact” or “DN” receptors.

5) Introduction. Please include sequence alignments of a few receptors in Figure 1 to show conservation of the salt-bridge across family and generality of the study. Also, please explicitly identify the residue position number of the acidic residue on M4 in the text.

Sequence alignments have been added to Figure 1 and the residue positions have been included in the main text.

6) Statements such as "These results localize the primary effect of the salt bridge disrupting mutation…" are model dependent, and the issue is with the word "primary". Which model do the authors have in mind? In the usual sequential model the open-closed behavior at saturating agonist is indeed due only to the final opening step; but is the change sufficient to explain (at least approximately) the shifted dose-response curve? Or is a substantial change in binding constants required as well? Similarly, the charge-reversal mutation DKKD is presented as nearly equivalent to the wildtype channel, but it is in fact a partial rescue, and the completeness of its rescue might best be evaluated with a model.

Agree. We now fit the data to the simple sequential model consisting of two equivalent binding steps followed by a gating step. Those fits are now displayed in Figures 2-4. Within this model, a change in the gating step predominantly accounts for the changes observed in the open probabilities of the DN mutant receptors. There is also a small decrease in the apparent dissociation constant for constructs with DN mutations in the β subunit, though these changes in the binding step are small relative to the change in the gating step.

7) A related issue is the existence of a new, longer open time component at high agonist concentrations that is mentioned in the Results. Please show some data to give the reader an idea of its importance. Did this affect the saturating open probability calculations?

There is a small component of slightly longer open times in the DN mutants. The fact that this component persisted even at high concentrations, suggests a second open state in the mutant receptor even when it is doubly occupied. However, this second open state has no direct impact on our determinations of saturating open probability. The reason is that open probability is a model-independent quantity; P_open_ is the ratio of open dwell times divided by the sum of open plus closed dwell times. If this second open component were not present in the DN mutant data, the saturating P_open_ would decrease further. We do take your point however, that this limits the application of a general model for both the WT and mutant receptor. However, given the large change in saturating P_open_, the reduction in open dwell times, and the increase in inter-cluster closed dwell times, our conclusion that the DN mutation impairs channel gating is justified.

8) Fluctuations that are so slow (~1 ms) are unlikely to be arise from the breaking of a salt bridge that (one would expect) would leave a helix flopping in the breeze on a ~1 ns timescale. So there's something complicated going on here with slow rearrangements of the protein that have only small effects on the conductance. A discussion of the relation of these timescales to potential mechanisms would be helpful for many readers.

Agree. We have added a paragraph that discusses the expected time courses of noise generating processes (Discussion).